# Investigation of Morphological, Optical, and Dielectric Properties of RF Sputtered WO_x_ Thin Films for Optoelectronic Applications

**DOI:** 10.3390/nano12193467

**Published:** 2022-10-04

**Authors:** Samiya Mahjabin, Md. Mahfuzul Haque, K. Sobayel, Vidhya Selvanathan, M. S. Jamal, M. S. Bashar, Munira Sultana, Mohammad Ismail Hossain, Md. Shahiduzzaman, Merfat Algethami, Sami S. Alharthi, Nowshad Amin, Kamaruzzaman Sopian, Md. Akhtaruzzaman

**Affiliations:** 1Solar Energy Research Institute (SERI), Universiti Kebangsaan Malaysia (The National University of Malaysia), Bangi 43600, Malaysia; 2Bangladesh Council of Scientific and Industrial Research, Dhaka 1205, Bangladesh; 3Department of Electrical and Computer Engineering, University of California, Davis, CA 95616, USA; 4Nanomaterials Research Institute (NanoMaRi), Kanazawa University, Kakuma, Kanazawa 920-1192, Japan; 5Department of Physics, Faculty of Science, Taif University, P.O. Box 11099, Taif 21944, Saudi Arabia; 6Institute of Sustainable Energy, Universiti Tenaga Nasional (The National Energy University), Jalan IKRAM-UNITEN, Kajang 43000, Malaysia; 7Graduate School of Pure and Applied Sciences, University of Tsukuba, Tsukuba 305-8573, Ibaraki, Japan

**Keywords:** tungsten oxide, sputtering, RF magnetron sputtering, metal oxide, photovoltaic, optoelectronics, perovskite solar cell, electron transport layer

## Abstract

Tungsten oxide (WO_x_) thin films were synthesized through the RF magnetron sputtering method by varying the sputtering power from 30 W to 80 W. Different investigations have been conducted to evaluate the variation in different morphological, optical, and dielectric properties with the sputtering power and prove the possibility of using WO_x_ in optoelectronic applications. An Energy Dispersive X-ray (EDX), stylus profilometer, and atomic force microscope (AFM) have been used to investigate the dependency of morphological properties on sputtering power. Transmittance, absorbance, and reflectance of the films, investigated by Ultraviolet-Visible (UV-Vis) spectroscopy, have allowed for further determination of some necessary parameters, such as absorption coefficient, penetration depth, optical band energy gap, refractive index, extinction coefficient, dielectric parameters, a few types of loss parameters, etc. Variations in these parameters with the incident light spectrum have been closely analyzed. Some important parameters such as transmittance (above 80%), optical band energy gap (~3.7 eV), and refractive index (~2) ensure that as-grown WO_x_ films can be used in some optoelectronic applications, mainly in photovoltaic research. Furthermore, strong dependencies of all evaluated parameters on the sputtering power were found, which are to be of great use for developing the films with the required properties.

## 1. Introduction

Transition metal oxides (TMOs) have fascinated researchers for their prospective applications. Among different TMOs, tungsten oxide (WO_x_) is a promising material for environmental and energy applications [1]. Due to its outstanding optochromic, electrochromic, and gasochromic properties, it has been used to produce numerous devices such as electrochromic ‘smart’ windows, flat-panel displays, humidity, and temperature sensors, gas sensors, optical modulation devices, and field emission devices [2]. Moreover, the potential uses of WO_x_ in many photovoltaic applications such as dye-sensitized solar cells [3], amorphous silicon solar cells [4], etc., have motivated researchers for investigating its usage in perovskite solar cells (PSCs) [5].

PSCs have progressed more rapidly in power conversion efficiency than any other photovoltaic (PV) technology in history, reaching above 25% in recent years [6,7,8,9]. Generally, PSCs are designed with the transparent conducting oxide (TCO)/electron transport layer (ETL)/perovskite absorber layer (PAL)/hole transport layer (HTL)/metal electrode configuration. The ETL that extracts electrons generated by the incident light in the PAL and transports them to electrodes plays a vital role not only in the performance but also in the degradation and stability of PSCs [10].

As ETL, n-type metal oxides such as titanium dioxide (TiO_2_), zinc oxide (ZnO), and tin dioxide (SnO_2_) have been widely used in PSCs. However, TiO_2_ mostly used ETL and affects the efficiency and stability of PSCs because of its highly dense trap states, degradation due to light, hysteresis behavior in current-voltage properties, non-radiative loss due to oxygen vacancy formation, non-radiative loss due to oxygen vacancy formation, and interface charge accumulation [10,11]. In addition, TiO_2_ exhibits lower electron mobility of ~1 cm^2^/Vs [12] that lowers electrical conductivity proportionally. For achieving high electrical conductivity of this material, a high sintering temperature is required, which is unsuitable for economical processing [13]. Likewise, SnO_2_ requires post-annealing at temperatures above 180 °C. Ali et al. reported that annealing at high temperatures not only adds complexity and expense to produce PSCs but also prevents developing flexible PSCs further [14]. On the other hand, though ZnO can be prepared through a simple hydrothermal method and it possesses high electron mobility, still, there are a few limitations for ZnO to be used in PSCs as ETL [13]. According to Wang et al., weak acids or bases could react with ZnO [15]. This affects its surface morphology as well as optoelectrical properties [10]. Additionally, the PAL may be decomposed to PbI_2_ upon thermal annealing on the ZnO surface because of the reaction between unconfined OH− ions from ZnO and the component CH_3_NH_3_I [16]. These demerits of TiO_2_, SnO_2_, and ZnO hamper PSCs’ efficiency and stability.

Alternatively, WO_x_ has a high electron mobility of 10–20 cm^2^/Vs that enables efficient transportation of photo-generated electrons [17]. The optical band energy gap of WO_x_ ranges from 2.7 to 3.9 eV depending on different structures [18]. WO_x_ is highly transmissible in the visible light range of the solar spectrum, with a relatively low refractive index (2.1) [19]. Band alignment with the perovskite absorber layer is comparatively good for WO_x_ [17]. Low material cost and low-temperature processing technology have made WO_x_ more promising [5,19]. Moreover, it has high chemical stability against acid/base solution, high stability against moisture [16], and enough tolerance to oxygen vacancies under the illumination of the ultraviolet region, which confirm the great opportunities in extending PSCs’ lifetime. In addition, the uniformly dense WO_x_ film prevents electron–hole pair recombination at the PAL-TCO interface by avoiding the shunt pathways between them.

Numerous methods for the synthesis of WO_x_ thin films have been reported, including sol-gel [20], chemical vapor deposition [21], electron-beam evaporation [18], etc. The solution-based spin-coating processing method has been commonly used to synthesize WO_x_ film. However, post-annealing treatment is necessary for the films developed by this method for eliminating the solvent and for decomposing the precursors into WO_x_, which is a major disadvantage [14]. Maintaining the uniform film quality over a large area of the substrate is another problem of this method. On the other hand, sputtering is a fantastic method that facilitates enough accuracy in film thickness. Moreover, elemental composition and impurity in the films can be controlled up to the desired level by this method [22]. Therefore, this method may be considered as the best method for growing WO_x_ thin films.

Not enough studies have been published in which WO_x_ was deposited through sputtering. Few of these studies have focused on dc magnetron sputtering [23], and others have used RF magnetron sputtering [24,25,26]. In some research, reactive sputtering has been used to deposit WO_x_ thin films [27]. Most of this research has studied the electrochromic properties [23,25] and electrochemical properties [26] of sputtered WO_x_. Few reports have studied optical properties for proposing the WO_x_ for electrochromic devices [23]. Though few research articles have been published that focus on the RF magnetron sputtering, none of them has deposited films by varying sputtering power. In this research, RF magnetron sputtering has been used to develop WO_x_ thin films. At the target of tungsten trioxide (WO_3_), applied RF sputtering power was varied for developing different films at room temperature. To the best of our knowledge, no research has been conducted on the study of optical properties of RF magnetron sputtered WO_x_ developed at varying RF power for proposing them for optoelectronic devices. In this study, the structural, morphological, optical, and dielectric properties of sputtered WO_x_ films have been analyzed by X-ray diffraction (XRD), Energy Dispersive X-ray (EDX) spectrometer, Dektak Profilometer, Atomic Force Microscope (AFM), UV-Vis spectrometer, etc., for verifying their uses in optoelectronic research. Due to the changes in sputtering power, clear variation in grain size was observed, and the dependency of several films’ properties on grain size was analyzed, which had not been explained so widely for sputtered WO_x_ films earlier. The effects of the wavelengths of incident light on those properties were also analyzed.

## 2. Methodology

### 2.1. Deposition of WO_x_ Thin Film

All the glass substrates needed to be cleaned before entering into the sputtering chamber. This cleaning process was executed in an ultrasonic bath by the sequential flow of soap water–deionized water–methanol–acetone–deionized water. Then, N_2_ gas was flown to dry the substrates. Finally, plasma cleaning was performed to remove organic contaminants. The WO_3_ sputtering target of Kurt J. Lesker Company (Jefferson Hills, PA, USA) was used. The substrate remained 11 cm apart from this target. A molecular turbo pump lowered the base pressure of the deposition chamber close to 2.4 × 10^−6^ Torr at first. After starting the deposition, the flow of Ar was maintained in such a way that the working pressure remained fixed at 15 m Torr in the chamber during the whole deposition process. Sputtering power was changed from 30 W to 80 W to develop different WO_x_ films by maintaining the deposition time for 30 min.

### 2.2. Characterization

The as-deposited WO_x_ films were examined by an XRD instrument (BRUKER, Germany) named BRUKER D8 ADVANCE to obtain the structural properties. XRD patterns were recorded in the 2θ range of 20° to 60° for the step size of 0.02° by Cu Kα radiation of wavelength 1.5406 Å. For measuring the thickness of the films, a Dektak XT Profilometer (BRUKER, Germany) was used. EDX spectroscopy of Zeiss EVO 18 Research instrument (ZEISS, Jena, Germany) was used to reveal the elemental composition of as-grown films. Surface topographies of the films were carried out by NANOSURF C3000 CONTROLLER AFM instrument (Nanosurf, Liestal, Switzerland), and by using these results, the grain size and roughness of the films were measured. A UH4150 UV-VIS-NIR Spectrophotometer instrument (Hitachi, Japan.) measured the optical transmission, absorbance, and reflectance spectra of the films due to the light of a visible and near-visible range. From these spectra, the optical band energy gap, refractive index, and a few other parameters that are important for optoelectronic research were determined.

## 3. Results and Discussions

### 3.1. Morphological Characterization

EDX results of WO_x_ thin films, developed at various sputtering powers, are shown in Appendix A. According to these figures, it is clearly found that the weight % of tungsten increases from 68.05% to 77.24% in the developed WO_x_ films, with the increment of sputtering power from 30 W to 80 W. This incremental behavior of tungsten concentration is tabulated in Table 1 and clearly shown in Figure 1a.

Several values of thickness of WO_x_ films at different positions were measured by the Dektak XT Profilometer instrument). Average thicknesses of the films, developed at different sputtering powers from 30 W to 80 W, were measured between 55.334 nm and 113.2 nm, which are tabulated in Table 1. The thickness of the films increases linearly with the sputtering powers, as illustrated in Figure 1b. The incremental trend of thickness may be described by the kinetic energy’s gaining of the positive ions in the plasma. As the sputtering power is increased, i.e., a higher negative voltage is applied to the cathode (target); positively charged ions are highly attracted by the target and gain higher kinetic energy [28]. As a result, these ions bombard the target of WO_3_ with higher velocity, and a higher number of sputtered atoms are released in the sputtering chamber, which causes the higher growth rate of the film [29]. Therefore, the sputtering power has great control over the surface morphology, which has noteworthy effects on the functional properties of thin films [30].

From AFM analysis, shown in Figure 2a–f, it has been observed that the average and the root mean square (RMS) value of surface roughness vary from 0.597 nm to 3.253 nm and 0.758 nm to 3.913 nm, respectively, as the sputtering power increases from 30 W to 80 W. This incremental trend in surface roughness of the WO_x_ films with the increasing value sputtering power is clearly shown in Figure 2g. The higher value of roughness indicates the presence of more pinholes or cracks. As smooth, uniform, and pinhole-free ETL is a vital prerequisite for highly efficient and stable PSCs [19], the films of WO_x_ developed at lower sputtering power are more suitable as ETL than the films developed at higher power. From the surface scan of two-dimensional (2D) AFM images of Appendix A, grain sizes of the developed films have been evaluated, as grain size is an important factor of the surface morphology of the films, and it is used to interpret the variation in optical properties of the films. For the developed film, the average grain size increases from 46.36 nm to 67.74 nm when sputtering power rises from 30 W to 80 W, which is clearly illustrated in Figure 2h. The larger grain size confirms the minimization of the grain boundary in the sample and, as a result, interfacial scattering losses are reduced [22]. All morphological properties which were evaluated are tabulated in Table 1.

### 3.2. Structural Characterization

The XRD measurement was carried out for studying the structural properties that include the crystalline states of all synthesized WO_x_ films. The XRD patterns for those WO_x_ films, deposited at various sputtering powers, are illustrated in Figure 3. All films show the amorphous state. This finding can be explained by the temperature. From different literature, it has been found that the WO_x_ films transfer to the crystalline state from the amorphous at 500 °C or higher [15]. Up to 385 °C, the phase of WO_3_ films remains amorphous, begins crystallizing at 390 °C, and, finally, at 450 °C, films become completely crystallized [31]. Moreover, the energy of the bombarding ions was small at low sputtering power. Therefore, the surface energy of the atomic grouping along the planes is lower, for which films show very poor crystallinity, i.e., are amorphous at room temperature [32].

### 3.3. Optical Properties

#### 3.3.1. Optical Transmittance, Reflectance, Absorbance, and Absorption Coefficient

To understand optoelectronic properties and to determine some optical parameters, optical transmittance, absorbance, and reflectance studies are essential. Figure 4a–c show the optical transmittance, reflectance, and absorbance spectra for WO_x_ films that were produced at various sputtering powers. From the graph of Figure 4a, the calculated average value of the transmittance over 400 nm to 1050 nm light wavelengths decreases gradually from 92% to 83.5% with the increment in sputtering power. The decremental behavior of transmittance can be explained with the help of thickness and grain size. As higher power produces thicker films, it is obvious that the transmittance decreases gradually with the incremental thickness of the films. On the other hand, high sputtering power produces higher grain-sized films, which show lower transmittance due to Rayleigh scattering [33]. The high value of transmittance suggests that developed WO_x_ films are appropriate for use in optoelectronic devices that require high transmittance. For example, deposited films are suitable to be used in PSCs as ETL [34].

From the reflectance spectra of Figure 4b, it is observed that films grown with higher sputtering power show more reflectivity for the light above 650 nm wavelength. This incremental behavior may be described by the scattering of lights due to the increment of surface roughness and grain size [35].

Though the absorbance spectra of Figure 4c do not show a significant pattern for the light wavelengths below 650 nm, above this wavelength, higher sputtering powered films show higher absorbance (A). Additionally, the calculated average value of A over 400 nm to 1050 nm light wavelengths increases gradually with the increment of sputtering power. It means that higher thick films show higher absorbance. This incremental behavior of A can be explained by the plasmon energy of the deposited WO_x_. A higher absorbance peak represents a higher plasmon intensity due to the increment of the carrier concentration. In this study, a higher value of tungsten (as weight %) in the films, deposited at higher sputtering power, ensures that less oxygen (O) is incorporated in the films, leading to an increase in carrier concentration. As a result, the plasmon intensity is higher for the films which are deposited at higher sputtering power, and higher absorbance is observed [36]. On the other hand, according to this figure, the absorption edges, where the difference in energy between the conduction band and the valence band is minimal corresponding to the beginning of an electronic transition between these bands, commence near 400 nm. These absorption edges near this wavelength of light correspond to the excitonic bandgap of WO_x_ that confirms semiconductor-type behavior [19,37].

The values of absorption coefficient (α) were determined for all films from absorbance (*A*) and thickness (*d*) of the films by using the relation [38]:(1)$$\alpha =\frac{2.303\;A}{d}$$

α plays a vital role in determining some parameters of thin films, such as optical energy bandgap, Urbach energy, extinction coefficient, etc., which are very important for explaining some necessary characteristics of the films. Figure 4d shows the variation in absorption coefficients for all films with different wavelengths of light. For the lights with a higher wavelength than 650 nm, α increases with the increment in sputtering power.

#### 3.3.2. Penetration Depth

Penetration depth (δ) is another important optical parameter that is also used to characterize the light absorption within thin films. The thickness, at which the light intensity drops to 37% of the initial intensity at the surface, is known as penetration depth. It depends on the structure of the field and incident photon energy. Since the conductivity in semiconductors is strongly dependent on the optical energy bandgap, the optical properties can be related to the penetration depth (δ) [19]. By using the absorption coefficient (α), δ can be determined from the following relation [19]:(2)$$\delta =\frac{1}{\alpha }$$

Figure 5a shows the dependence of δ on light wavelength for all WO_x_ films developed at different sputtering power. It is revealed from the figure that δ decreases with the increment of sputtering power and increases with the increment of light wavelength. It is to be observed that the values of δ are higher for the light wavelengths near 400 nm. This behavior can be explained by the transmittance and absorbance spectra, as δ depends strongly upon T and A values [39]. Observing the T and A behavior of the films, we can say that the absorption edges are found near the 400 nm wavelength of light, which corresponds to the energy bandgap of the material. Due to the absorption edges’ sudden decrement of α near that wavelength of light, this confirms the higher values of δ. This sudden decrement may occur due to electron coupling [39].

#### 3.3.3. Energy Bandgap

Among different optical properties of a material, the bandgap energy (E_g_) is very significant because of its necessity to determine the range of photosensitivity of that material. E_g_ can be found by linearizing the Tauc’s equation that relates α and photon energy (hν) with E_g_, as [40]:αhν = A(hν − E_g_)*^n^*(3)
where A is a constant depending on the spectrum band edge, *n* has the value of 1/2, 3/2, 2, or 3 corresponding to the direct allowed, direct forbidden, indirect allowed, or indirect forbidden transitions, respectively [19,41]. The linear part of the curve that presents (αhν)^1/*n*^ vs. photon energy (hν) is extrapolated until it intersects the hν-axis, and the hν value of that intersection point represents the value of E_g_ of the respective film [42]. Since WO_X_ has a direct bandgap (*n* = 1/2), Tauc’s plots are just (αhν)^2^ vs. hν for the deposited films, which have been drawn in Figure 5b. Using the mentioned extrapolation method of Tauc’s plots, the values of E_g_ have been determined between 3.64 eV and 3.75 eV for different WO_X_ films, sputtered at different powers. Table 2 represents these E_g_ values for the films prepared at different sputtering powers. Red-shift is observed in the values of E_g_ of the films as the sputtering power increases. This is because of the decrement of oxygen vacancies due to the increment of tungsten, as weight % in the films sputtered at higher power [36]. The variation in E_g_ with the increment of sputtering power is opposite to the incremental behavior of the grain size, i.e., lower grain size leads to higher E_g_. The quantum size effect is the main reason behind this bandgap widening [43]. As the grain size becomes lower, quantum confinement leads E_g_ to be increased. In the amorphous films, higher grain size leads to the decrease in the defect state density, resulting in the reduction in energy band tailing. 

The values of E_g_ remain between the reported values of 3.14 eV [34] and 3.84 eV [17]. These values of bandgap energy confirm the suitability of using WO_x_ as ETL in PSCs [44].

#### 3.3.4. Urbach Energy

For the absorption coefficient $\;\alpha \leq 10^{4}\;{\mathrm{cm}}^{-1}$, an Urbach tail is usually found, in which α is dependent on the photon energy as [29]:(4)$$\ln\alpha =ln\;\alpha_{0}+\left(\frac{h\nu }{E_{u}}\right)$$
where *E_u_* is the Urbach energy. The Urbach energy, which is less than the optical energy band gap, is associated with the overall contributions of the thermal disorder as well as static disorder. The inherent structural disorder is the main reason behind the static disorder, while excitations of phonon modes cause the thermal disorder [45,46]. Thus, the Urbach effect is the outcome of the collective effect of different disorders, impurities, and electron–phonon interaction in the absorption processes [47,48]. By following the relation of Equation (4), ln α vs. photon energy (*hν*) graphs for all films have been plotted, and partial graphs of the absorption edge region are shown in Figure 5c, from which the slopes of the linear parts have been determined. The reciprocals of these slopes represent the values of Urbach energy, which were found between 283 and 422 meV for different WO_x_ films.

#### 3.3.5. Refractive Index and Extinction Coefficient

The refractive index (*n*) and other dispersion parameters perform a major role to determine the electronic properties of semiconductor materials used in optoelectronic devices [35]. *n* is related with the complex refractive index (*n*^*^) as in the following relation [18]:*n*^*^ = *n* + iK(5)
where the imaginary part K is known as the extinction coefficient that points to scattering and absorption-related optical loss. When light penetrates a medium, some portion of the light is lost because of scattering and absorption by the medium and K measures that lost part of light [49]. The following relations guide to calculating the value of *n* and K [18]:(6)$$n=\frac{1+R}{1-R}+\sqrt{\frac{4R}{{\left(1-R\right)}^{2}}-K^{2}}$$
(7)$$K=\frac{\lambda \alpha }{4\pi }$$
where *R* is reflectance. Figure 6a shows the variation in *n* with the photon wavelength for the WOx films developed with different sputtering powers. It is found that *n* has a greater value (up to 2.4) around the absorption edge. In the ultraviolet region of light, *n* decreases most rapidly with the increment of the light wavelength representing the dispersive behavior of light that is consistent with many semiconducting characteristics [35]. The most important observation found from the figure is the incremental behavior of *n* above the light of 650 nm wavelengths with the increment of sputtering power, i.e., films prepared with higher sputtering power show a higher refractive index. This changing behavior may be described by means of polarizability [50]. It has been found from the EDX results that higher sputtering power ensures a higher concentration of tungsten (W) and lower concentration of oxygen (O) in the developed films. As we know that the atomic radius of W (2.10 Å) is greater than the atomic radius of O (1.52 Å), the higher concentration of W in the sputtered films confirms the domination of higher radius atoms over lower radius atoms. As a result, films developed with higher sputtering power show higher polarizability and, consequently, the refractive index (*n*) raises. This incremental behavior of *n* may also be described from the increment of the atomic packing density of the films. An increment of the concentration of atoms with a higher atomic radius causes the increment of packing density and, consequently, the velocity of light becomes lower, which increases *n* [51,52,53]. From a close observation, it can be found that the less thick films show a peak ne 1 ar 400 nm. This trend also matches with the R spectra, as *n* is calculated directly from the R values. Near this wavelength of light, absorbance edges are found where A falls drastically at a high rate. Due to the lowest value of A there, T and R tend to show the maximum values. Absorbance edges are related to the band gaps of the films, which are found closely equal to the photon energy of the 400 nm light. In brief, it can be expressed that the peak value of *n* can be said to be correlated with Eg. There is a red-shift of peaks of *n* that can also be correlated with the red-shift of the bandgap energy with the increasing value of sputtering power, that is, for the increment of free carriers, leading to the decrease in O vacancies. The derived values of *n* for the developed WO_x_ films are between 1.6 and 2.3, which are matched with the reported values of 1.6 to 2.2 [19] and 1.9 to 2.15 [34]. The developed WO_x_ films’ refractive indices are closely matchable with the refractive index of the perovskite absorber layer. Thus, these films are perfect for use in PSCs as ETL [54].

Figure 6b illustrates the changes in K for the different WO_x_ films, developed by different sputtering powers with respect to the variation in light wavelengths. Though there was no general trend of the variation in K values for the light of wavelengths below 650 nm, K slightly increases above this wavelength as the sputtering power is increased. Evidently, K has lower values (0–0.35) because of the surface smoothness of WO_x_ thin film [30]. These lower values indicate the lesser loss of light energy due to absorption and scattering [19]. This reason makes WO_x_ thin films suitable for photovoltaic applications [35]. 

#### 3.3.6. Optical Conductivity

Characterizing the optical response of a material is very important for using it in optoelectronic research. To study this response conveniently, optical conductivity (*σ_opt_*) is a significant property, which represents charge carriers’ conductivity through the material due to the optical excitation [55]. *σ_opt_* can be determined by *n*, *α*, and velocity of light (*c*) using the following relation [56]:(8)$$\sigma_{opt\;}=\frac{\alpha nc}{4\pi }$$

The variations in *σ_opt_* for developed WO_x_ films with respect to the light wavelengths are shown in Figure 6c.

#### 3.3.7. Dielectric Constant

The dielectric constant (*ε*) is an important parameter because this is associated with the fundamental electronic transition and excitonic characteristics in semiconductors [57,58]. The materials having a higher value of ε are enriched with the intrinsic charge carrier [48]. Furthermore, some optically important parameters such as dielectric loss factor (tan δ) or dissipation factor, volume and surface energy loss functions, etc., can be derived using *ε* [49]. Therefore, as a whole, *ε* plays a noteworthy role in the design, optimization, and evaluation of optoelectronic devices [59]. *ε* constitutes real (*ε_r_*) and imaginary (*ε_i_*) parts as [19]:(9)$$\varepsilon =\varepsilon_{r}+i\varepsilon_{i}$$

*ε_r_* and *ε_i_* are related with *n* and *K* according to the relations [32]:(10)$$\varepsilon_{r}=n^{2}-K^{2}$$
(11)$$\varepsilon_{i}=2nK$$

Generally, *ε_r_* is associated with the dispersion of the electromagnetic waves through the studied material. The reduction in light propagation speed through a material depends on the value of *ε_r_* of that material [20]. On the other hand, the absorbed energy from an electric field created by moving the dipole depends on *ε_i_*. Moreover, it provides the disruptive rate measurement of the wave in the material [49]. Variations in *ε_r_* and *ε_i_* with respect to the light wavelength for as-grown WO_x_ films are illustrated in Figure 7a and Figure 7b, respectively. It is clearly found from these figures that *ε_r_* and *ε_i_* values increase with the increment of sputtering power for the light above 650 nm. Therefore, the increasing sputtering power tends to raise the energy destructive and dissipative rates of the incident photons on WO_x_ films [35].

### 3.4. Loss Parameters

#### 3.4.1. Loss Factor

The loss factor (tanδ), sometimes called the dielectric loss factor or dissipation factor, is another important parameter for investigating the structure and defects in solids [60]. It is correlated to the loss of energy due to the electromagnetic field-absorbing ability of the material [61]. Phase difference due to this energy loss is represented by this loss factor (tanδ), that can be calculated by the following relation [19,60]:(12)$$\tan\delta =\frac{\varepsilon_{i}}{\varepsilon_{r}}$$

Figure 8a depicts the variation in loss factor for different WO_x_ films on light wavelengths. For the light above 650 nm wavelength, it is observed that films developed with higher sputtering power have higher values of loss factor.

#### 3.4.2. Volume Energy Loss Function and Surface Energy Loss Function

Volume energy loss function (VELF) and surface energy loss function (SELF) are other important parameters that are related proportionally to the characteristics of energy loss of highly speedy and energetic electrons traveling along the material [62]. Energy loss due to the passing of electrons through the material and along the surface area was measured by the VELF and SELF, respectively. These loss functions are associated to the optical properties of the films through the dielectric constants (*ε_r_* and *ε_i_*) according to the following relations [62]: (13)$$VELF=\frac{\varepsilon_{i}}{\varepsilon_{r}^{2}-\varepsilon_{i}^{2}}$$
(14)$$SELF=\frac{\varepsilon_{i}}{{\left(\varepsilon_{r}+1\right)}^{2}+\varepsilon_{i}^{2}}$$

Figure 8b,c show the dependence of VELF and SELF, respectively, for as-grown films of WO_x_ upon the wavelength of incident light. Both curves show almost similar behavior.

With the increment of sputtering power, films face higher energy loss. By comparing these two types of graphs, it is found that SELF is a bit more prominent than VELF. Consequently, the free charge carriers suffer more energy loss at the time of traveling along the surface than the energy loss due to the collisions within the material [63]. Low volume energy loss compared to the surface energy loss is a promising factor for choosing the developed WO_x_ as the ETL in the PSCs, because free carriers in ETL have to pass through the material, not along the surface.

#### 3.4.3. Reflection Loss

Loss due to reflection is unwanted for most of the optoelectronic applications, e.g. in solar cell application, most of the light should be transmitted through the ETL. The reflection loss factor (*R_L_*) represents this loss, which can be calculated by using the refractive index (*n*) according to the following formula [49]:(15)$$R_{L}=\frac{n^{2}-1}{n^{2\;}+2}$$

The dependence of the *R_L_* of the developed films upon the wavelength of incident light is presented in Figure 8d. Films that were synthesized at a higher power face higher reflection loss, which is clearly observed from the values of *R_L_* for the light above the wavelength of 650 nm.

By observing the variation in the above-mentioned loss parameters, it can be concluded that the thick films (deposited at higher sputtering power) experience higher optical losses. This is not desirable for those optoelectronics devices in which higher intensity of the transmitted light through the WO_x_ films is important. Thus, WO_x_ films having the lowest thickness are expected to be developed. However, for this experiment, among films developed at 30 W and 40 W sputtering power, those that are less thick show some sort of instability in the T, A, and R spectra. Therefore, considering all the variations, films developed at 50 W sputtering power can be considered as optimum, where losses hamper the device’s performance, such as using WO_x_ as ETL in nip structured PSCs.

## 4. Conclusions

The morphological, optical, and dielectric properties of WO_x_ films that were developed under different sputtering powers ranging from 30 W to 80 W were investigated. The thickness, grain size, and roughness of the films were raised with the increment of sputtering power. Though higher sputtering power decreased the transparency of the films, highly transparent films were developed. The absorption edge shifted to the low energy side with the increment in sputtering power; as a result, the optical energy band gap decreased from 3.75 eV to 3.64 eV with the increase in sputtering power. Some important parameters such as refractive index, extinction coefficient, optical conductivity, real and complex dielectric constants, some losses, etc., were evaluated, and their dependency on sputtering power and light wavelengths was investigated. It has been found that sputtering power affects those important parameters of the films significantly. This experiment opens the gateway of controlling the mentioned properties by maintaining the sputtering power for a fixed time, which leads to research and development of WO_x_ films with optimum properties.

## Figures and Tables

**Figure 1 nanomaterials-12-03467-f001:**
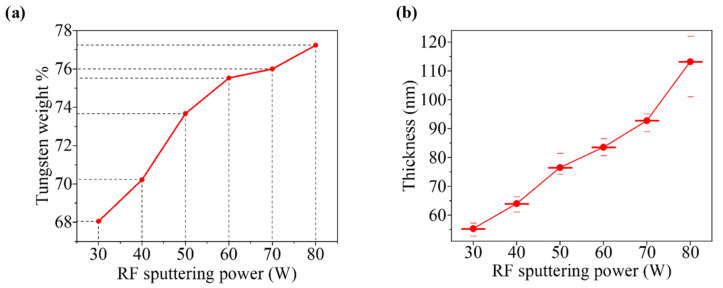
Variation of: (**a**) weight % of W and (**b**) thickness due to the RF sputtering power.

**Figure 2 nanomaterials-12-03467-f002:**
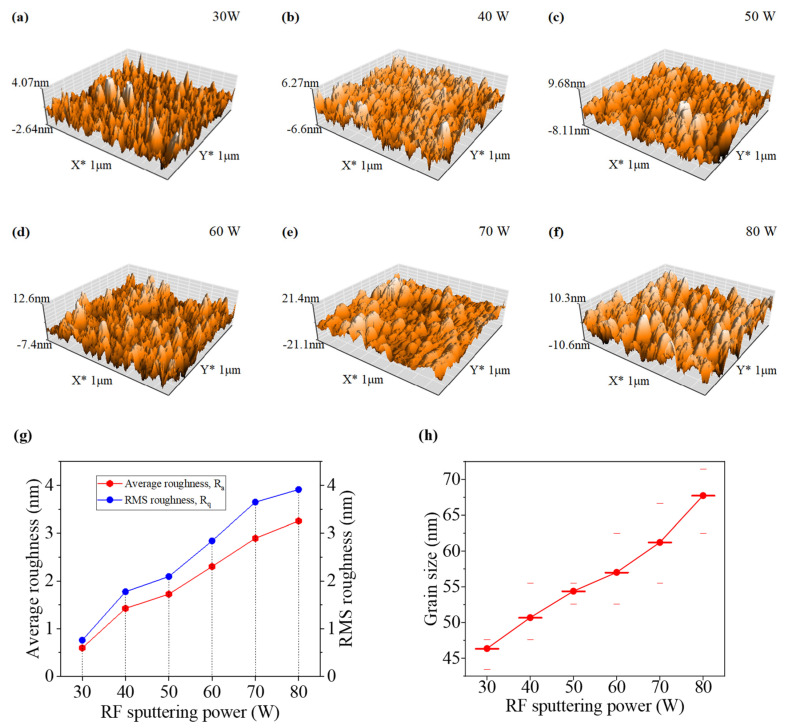
3D AFM images of the WO_x_ thin films, deposited at different sputtering powers: (**a**) 30 W, (**b**) 40 W, (**c**) 50 W, (**d**) 60 W, (**e**) 70 W, (**f**) 80 W. (**g**) Roughness variation and (**h**) grain size variation due to different RF sputtering powers.

**Figure 3 nanomaterials-12-03467-f003:**
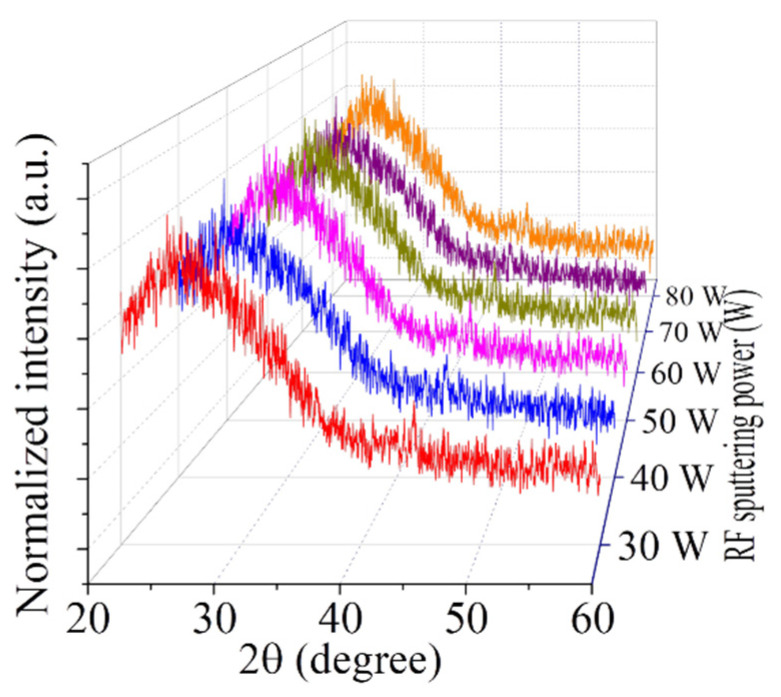
XRD patterns of the WO_x_ films grown at different RF sputtering powers.

**Figure 4 nanomaterials-12-03467-f004:**
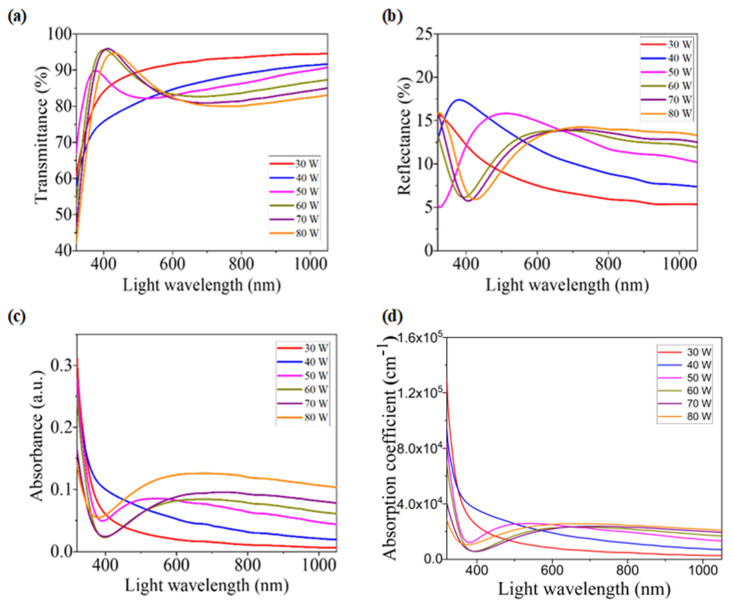
(**a**) Transmittance, (**b**) reflectance, (**c**) absorbance, and (**d**) absorption coefficient spectra for WO_x_ films developed at various RF sputtering powers.

**Figure 5 nanomaterials-12-03467-f005:**
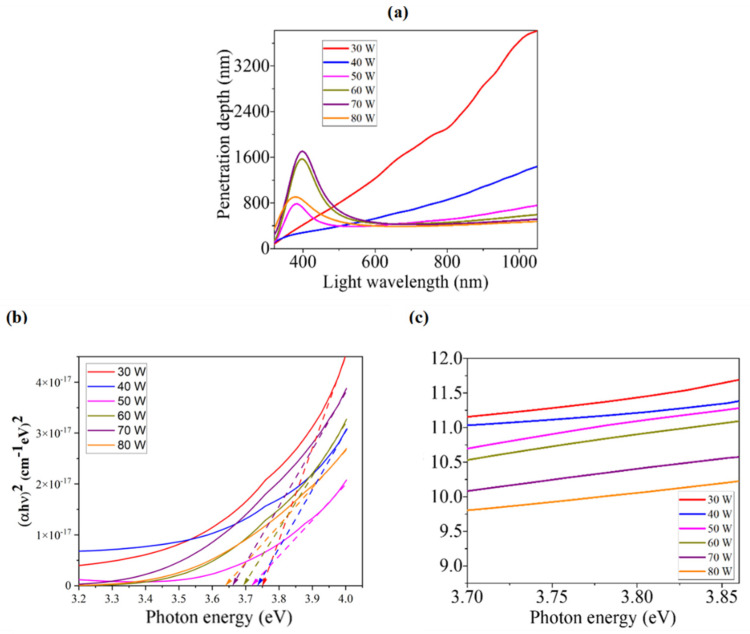
(**a**) Light wavelength dependence of penetration depth of as-grown films fabricated at different RF sputtering powers. Determination of: (**b**) energy bandgap using Tauc’s plot and (**c**) Urbach energy for different WOx films.

**Figure 6 nanomaterials-12-03467-f006:**
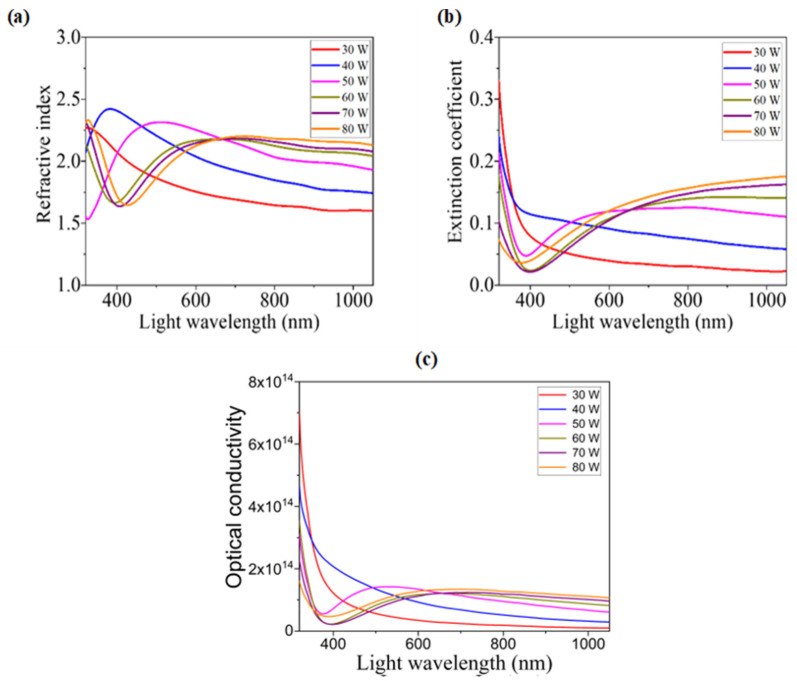
Variation in: (**a**) refractive index, (**b**) extinction coefficient, and (**c**) optical conductivity of different WO_x_ films due to the light wavelength.

**Figure 7 nanomaterials-12-03467-f007:**
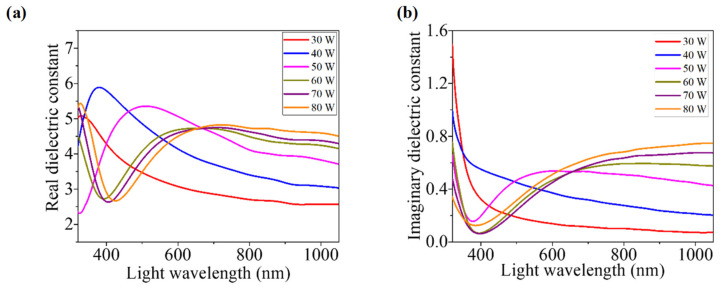
Light-dependent variation in: (**a**) real dielectric constant and (**b**) imaginary dielectric constant for different WO_x_ films developed at different powers.

**Figure 8 nanomaterials-12-03467-f008:**
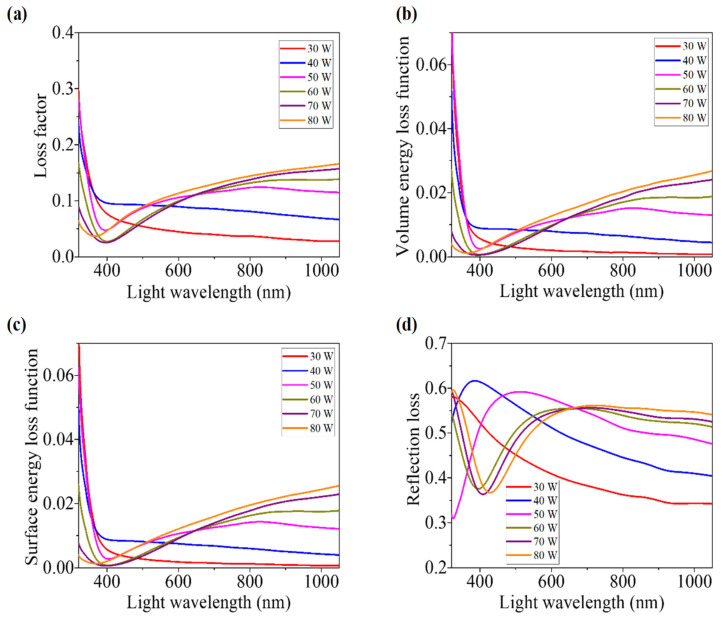
(**a**) Loss factor, (**b**) volume energy loss function, (**c**) surface energy loss function, and (**d**) reflection loss versus wavelengths of light for RF sputtering power-varied WO_x_ films.

**Table 1 nanomaterials-12-03467-t001:** Morphological properties of WO_x_ films.

RF Sputtering Power (W)	Weight % of W	Average Thickness (nm)	Average Grain Size (nm)	Average Roughness R_a_ (nm)	RMS Roughness R_q_ (nm)
30	68.05	55.334	46.36	0.597	0.758
40	70.22	64.006	50.685	1.424	1.775
50	73.76	76.508	54.386	1.724	2.093
60	75.53	83.55	57.013	2.301	2.839
70	76.01	92.767	61.21	2.893	3.648
80	77.24	113.2	67.74	3.253	3.913

**Table 2 nanomaterials-12-03467-t002:** Energy bandgaps for different WO_x_ films.

RF Sputtering Power (W)	Energy Band Gap E_g_ (eV)
30	3.751
40	3.737
50	3.72
60	3.695
70	3.66
80	3.642

## Data Availability

The data presented in this study are available on request from the corresponding author.

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
