# Peer review of "Investigation of Morphological, Optical, and Dielectric Properties of RF Sputtered WOx Thin Films for Optoelectronic Applications"

_nanomaterials, 2022, doi:10.3390/nano12193467_

Round 1
Reviewer 1 Report
The authors have synthesised several thickness tungsten oxide thin films using RF magnetron sputtering. The films were grown under different conditions and the physical properties were systematically characterised. AFM technique was used to study the morphology of the film, but the optimum roughness needed for the device fabrication was not explained. The as-grown films shows amorphous nature, however they mentioned other groups annealing results but, they didn't anneal their films.
The optical characterisation studies shows a clear difference with respect to the film thickness except 40W growth film. The penetration depth increases around 400nm for the thick film, again didn't explain the reason. The loss factors are important parameter for any applications, they presented the data, however it was not explained in detail.
Over all the article is well written with lots of references. The results were not discussed enough to arrive any meaning full conclusion.
I suggest the authors to consider the following points to improve the manuscript.
1. The nature of the as-grown films are amorphous and the authors knows the annealing conditions to improve the crystallinity, so I suggest to anneal the films and improve the crystallinity and repeat the measurement including optical studies.
2. Explain the reason for higher penetration depth for the thick films around 400nm?
3. Why the low power sputtered film shows higher refractive index around 400nm?
4. The loss factors are very important parameter for the application, Could you explain the optimum thickness suitable for the device fabrication.
Author Response
Responses to the Reviewer #1
Comment # 1: The Introduction part is lack of novelty statement. Since this work focuses on preparation and characterization, in my opinion, the first half in Page 2 is redundant. Does the proposed fabrication method represent a big progress in the preparation of WOx films? Have the similar characterizations of sputtered WOx films been reported? Why the characterizations in this paper are important for the future optoelectronic applications? These questions are more important.
Authors’ Response: Thank you for mentioning this important issue. We added to the first half of page 2 for providing the other preparation methods of WOx films so that we can say that the sputtering is a nice one. To support this para and to mention your other comments, we have rewritten the lines, which you will find in the last part of the introduction (line no. 100 to 107).
Comment # 2: In table 1, morphological properties of WOx films are concluded. It seems like all factors change with sputtering power synchronously. Is that possible to realize independently control? For example, synthesize a WOx film with large weight of W but small roughness.
Authors’ Response: Thank you very much for raising the important question. We have focused only on sputtering power variation in this study. So, all factors are changed with the sputtering power synchronously, as you have mentioned. But these parameters can be controlled independently. By maintaining higher sputtering power, a large weight % of W can be found in a film. We can control the films’ roughness by lowering the time of deposition. That means a lower time of sputtering means a lower number of bombardments and the roughness will be lower. On the other hand, by increasing the substrate to the target distance, roughness can also be lowered without lowering the time of deposition.
Comment # 3: In Page 7, the discussion on absorbance spectra is unconvincing. Many variables exist and contribute to absorbance via different mechanisms. It would be better to compare absorbance spectra of WOx films, for example, with different weights of W but fixed thickness. In addition, WOx with oxygen vacancies has been regarded as an important non-metallic plasmonic material ( J. Am. Chem. Soc. 2012, 134, 3995-3998), which strongly influences absorbance spectra.
Author’ Response: Thank you for mentioning the issue. We are grateful to you because of the reference you have suggested. We have rewritten the part of absorbance spectra by mentioning the issue of plasmon according to the reference. (line no. 230 to236)
Comment # 4: In Fig. 6a, the variation trend of refractive index with sputtering power is strange. How to explain the obvious red-shift from 30 W to 50 W and the stable curves of 60-80W? Mechanisms behind this phenomenon must be clarified. In my opinion, Weight% of W determines the free carrier concentration, which is important for calculating n and k based on Drude model.
Authors’ Response: Thank you very much for raising the critical issue. Red shift may be related to the red-shift of the band energy gap. We have tried to explain the issue in our revised manuscript. (Line no. 344 to 353, 288 to 290)

Reviewer 2 Report
In this paper, Samiya Mahjabin et al. studied the morphological and optical properties of WOx thin films synthesized by the RF magnetron sputtering. The authors have made detailed characterization and clear comparison of samples with different sputtering power. However, this paper is more like a technical instruction. A major revision is needed to improve the mechanism analysis and a logical connection from material properties to photovoltaic applications. Here are some detailed questions and concerns:
1. The Introduction part is lack of novelty statement. Since this work focuses on preparation and characterization, in my opinion, the first half in Page 2 is redundant. Does the proposed fabrication method represent a big progress in the preparation of WOx films? Have the similar characterizations of sputtered WOx films been reported? Why the characterizations in this paper are important for the future optoelectronic applications? These questions are more important.
2. In table 1, morphological properties of WOx films are concluded. It seems like all factors change with sputtering power synchronously. Is that possible to realize independently control? For example, synthesize a WOx film with large weight of W but small roughness.
3. In Page 7, the discussion on absorbance spectra is unconvincing. Many variables exist and contribute to absorbance via different mechanisms. It would be better to compare absorbance spectra of WOx films, for example, with different weights of W but fixed thickness. In addition, WOx with oxygen vacancies has been regarded as an important non-metallic plasmonic material ( J. Am. Chem. Soc. 2012, 134, 3995-3998), which strongly influence absorbance spectra.
4. In Fig. 6a, the variation trend of refractive index with sputtering power is strange. How to explain the obvious red-shift from 30 W to 50 W and the stable curves of 60-80W? Mechanisms behind this phenomenon must be clarified. In my opinion, Weight% of W determines the free carrier concentration, which is important for calculating n and k based on Drude model.
Author Response
Responses to the Reviewer #2
Comment # 1: The nature of the as-grown films are amorphous and the authors knows the annealing conditions to improve the crystallinity, so I suggest to anneal the films and improve the crystallinity and repeat the measurement including optical studies.
Authors’ Response: We are very much thankful for your nice suggestion. To do so, we have to deposit fresh films again maintaining the parameters and to anneal them. Then we have to characterize those films again. It will be quite time-consuming to complete these processes again. So, we apologize because of our inability to complete the whole process within a short period of time.
Besides, from the very beginning, we wanted to study the optical properties of RF sputtered as deposited WOx films only so that they can be verified for optoelectronic applications. Some researchers have already used amorphous WO3 as ETL in perovskite successfully. So, we planned to do some studies on as-deposited WO3 films only. As there was no study on verifying the suitability of using the WOx films in optoelectronic applications, which are prepared from variations of RF sputtering power. But, It would be really good if we could perform the annealing and further characteristics. Again, we apologize for our inability to do the study on annealed films within a short period of time.
Comment # 2: Explain the reason for higher penetration depth for the thick films around 400nm?
Authors’ Response: Thank you very much for the suggestion. We have revised the section penetration depth to address the issue you mentioned, which can be found in the revised manuscript (Line no. 259 to 268)
Comment # 3: Why the low power sputtered film shows higher refractive index around 400nm?
Authors’ Response: Thank you for mentioning the issue. We have tried to write down the probable reasons behind these trends in our revised manuscript. (Line no. 344 to 353)
Comment # 4: The loss factors are very important parameter for the application, Could you explain the optimum thickness suitable for the device fabrication.
Authors’ Response: Thank you very much for addressing the importance of loss factors. We have explained the optimum values for choosing in device fabrication based on the loss factors, which will be found in the last part of the loss factors (Line no. 447 to 455).

Reviewer 3 Report
This work describes morphological and optical studies of tungsten oxide thin films deposited with RF magnetron sputtering technique. Authors investigate several properties of thin films as a function of the sputtering power.
The manuscript is well written and structured, and the method and results will be of interest to other Nanomaterials readers. The paper is acceptable for publication after a few minor edits addressing the following comments:
I suggest shortening the abstract a little, avoiding such a detailed description of all the investigation techniques, and reporting only the salient information.
ln. 58. Give a definition of 'PV'
ln. 65. 'have been being widely used' -> 'have been widely used'
ln.118. I suggest adding commas: 'soap, water-deionized, ...'
Section 2.1 Deposition of WOx Thin Film: Please provide information on the deposition times.
Did you use the same deposition times with different sputtering power?
Figures S1 a-f: add units on axes. Also, could you comment on peaks not labeled near 1 keV?
ln. 142. here and later, I suggest using 'tungsten' rather than 'W', as you already extensively use 'W' as a power unit.
ln.150 'of 30 W to 80 W ' -> 'from 30 W to 80 W'
Figure 2. caption. I suggest breaking the sentence before '(g) roughness variation and '
Figure 5. Check y-axis units: 'V'->'eV'
Eq. 3: Seems wrong. It should be 'αhv = A(hv−Eg)^n'
ln.363 remove comma: 'It is clearly found from these figures that, εr ' -> 'It is clearly found from these figures that εr'
Eq. 12: it should be: 'epsilon_i / epsilon_r'
ln.420 'decreases' -> 'decreased'. (the sentence is written in past tense)
References: You should use ';' to separate author names, not ','
Author Response
Responses to the Reviewer #3
Comment # 1: I suggest shortening the abstract a little, avoiding such a detailed description of all the investigation techniques, and reporting only the salient information.
Authors’ Response: Thank you for mentioning this issue. We have shortened the abstract already as per your suggestion in the revised version.
Comment # 2: ln. 58. Give a definition of 'PV'
Authors’ Response: We have given the definition of PV in the revised manuscript (Line no xx to xx)
Comment # 3: ln. 65. 'have been being widely used' -> 'have been widely used'
Authors’ Response: We have corrected it in the revised manuscript (Line no 54)
Comment # 4: ln.118. I suggest adding commas: 'soap, water-deionized, ...'
Authors’ Response: Thank you for mentioning the issue, which can be misunderstood easily. That was a sequence of substrate cleaning. This will be like this: soap water - deionized water – methanol – acetone - deionized water. We have corrected the space so that it can be understood. (Line no 124 to 125)
Comment # 5: Section 2.1 Deposition of WOx Thin Film: Please provide information on the deposition times.
Did you use the same deposition times with different sputtering power?
Authors’ Response: Thank you for mentioning this issue. We have mentioned the deposition time in the revised manuscript (Line no 133). It is to be mentioned that we have used the same deposition time.
Comment # 6: Figures S1 a-f: add units on axes. Also, could you comment on peaks not labeled near 1 keV?
Authors’ Response: Thank you for mentioning the issue. We have added the unit of axes in the revised supplementary file as you mentioned.
Comment # 7: ln. 142. here and later, I suggest using 'tungsten' rather than 'W', as you already extensively use 'W' as a power unit.
Authors’ Response: We have corrected it in the revised manuscript (Line no 150 and 152).
Comment # 8: ln.150 'of 30 W to 80 W ' -> 'from 30 W to 80 W'
Authors’ Response: We have corrected it in the revised manuscript (Line no 158).
Comment # 9: Figure 2. caption. I suggest breaking the sentence before '(g) roughness variation and '
Authors’ Response: We have corrected it in the revised manuscript by putting a full stop (.) (Line no 188).
Comment # 10: Figure 5. Check y-axis units: 'V'->'eV'
Authors’ Response: We have corrected it in the revised manuscript (Line no 270).
Comment # 11: Eq. 3: Seems wrong. It should be 'αhv = A(hv−Eg)^n'
Authors’ Response: Thank you for addressing the major wrong. We have corrected it in the revised manuscript (Line no 279).
Comment # 12: ln.363 remove comma: 'It is clearly found from these figures that, εr ' -> 'It is clearly found from these figures that εr'
Authors’ Response: We have corrected it in the revised manuscript (Line no 397).
Comment # 13: Eq. 12: it should be: 'epsilon_i / epsilon_r'
Authors’ Response: We have corrected it in the revised manuscript (Line no 411).
Comment # 14: ln.420 'decreases' -> 'decreased'. (the sentence is written in past tense)
Authors’ Response: We have corrected it in the revised manuscript (Line no 464).
Comment # 15: References: You should use ';' to separate author names, not ','
Authors’ Response: Thank you for the suggestion. We are sorry for the issue. As we have used the citation style from the referencing software, these seem like that style. As from our previous experience, nanomaterials will update it according to their referencing style. So, we apologize to you for not changing the ‘,’ in this revised version. But after completing the technical review, we will update it as per your suggestion, if Nanomaterials’ referencing software does not correct it. We are feeling sorry again for not editing this in this revised version.

Round 2
Reviewer 2 Report
I would like to thank the authors for addressing all my comments.